Genetic structure of Trifolium pratense populations in a cityscape

Mollashahi Hassanali 1
Urbaniak Jacek 2
Szymura Tomasz H. 3
Szymura Magdalena magdalena.szymura@upwr.edu.pl 1
1 Institute of Agroecology and Plant Production, Wrocław University of Environmental and Life Sciences , Wrocław , Poland
2 Department of Botany and Plant Ecology, Wrocław University of Environmental and Life Sciences , Wrocław , Poland
3 Botanical Garden University of Wrocław , Wrocław , Poland
Sezgin Efe
Electronic publication date: 2023 Sep 5
Publication date: 2023
Volume: 11
Electronic Location ID: e15927
Received 2023 Feb 22; Accepted 2023 Jul 30
Copyright: ©2023 Mollashahi et al.
Copyright year: 2023
Copyright holder: Mollashahi et al.
License: This is an open access article distributed under the terms of the Creative Commons Attribution License, which permits unrestricted use, distribution, reproduction and adaptation in any medium and for any purpose provided that it is properly attributed. For attribution, the original author(s), title, publication source (PeerJ) and either DOI or URL of the article must be cited.
License URL: https://creativecommons.org/licenses/by/4.0/

Keywords: City grasslands, Genetic diversity, Isolation by distance, Isolation by resistance, Least cost patch analysis

Funding: UPWR 2.0: International and Interdisciplinary Programme of Development of Wrocław University of Environmental and Life Sciences European Social Fund under the Operational Program Knowledge Education Development POWR.03.05.00-00-Z062 The publication is financed by the project “UPWR 2.0: International and Interdisciplinary Programme of Development of Wrocław University of Environmental and Life Sciences”, and co-financed by the European Social Fund under the Operational Program Knowledge Education Development, under contract No. POWR.03.05.00-00-Z062/18 of June 4, 2019. The funders had no role in study design, data collection and analysis, decision to publish, or preparation of the manuscript.

==============================
Urban grasslands provide numerous ecosystem services, and their maintenance should be based on naturally regenerating plant populations. However, the urban environment is challenging for preserving viable populations, mostly because of their high fragmentation and small size, which can lead to genetic drift. We examined red clover (Trifolium pratense) in a medium-size city in Central Europe to test the cityscape effect on within- and among-population genetic diversity. We used eight inter-simple sequence repeat markers to examine the genetic structure of 16 populations, each represented by eight individuals. The isolation by resistance was analysed using a least cost patch approach, focusing on gene flow via pollinators. We found great variation among T. pratense populations, with no discernible geographic pattern in genetic diversity. We linked the diversity to the long history of the city and high stochasticity of land use changes that occurred with city development. In particular, we did not find that the Odra River (ca. 100 m wide) was a strong barrier to gene transfer. However, notable isolation was present due to resistance and distance, indicating that the populations are threatened by genetic drift. Therefore, gene movement between populations should be increased by appropriate management of urban green areas. We also found that small urban grassland (UG) patches with small populations can still hold rare alleles which significantly contribute to the overall genetic variation of T. pratense in the city.

Introduction

Urbanisation has shown almost exponential growth since the end of the 19th century (Champion, 2001), and further increases in urban populations are predicted (Zhang, 2008). Surprisingly, cities can play a critical role for native biodiversity, which can be maintained through planning, conservation, and management of urban green spaces (Angold et al., 2006; Aronson et al., 2017a; Aronson et al., 2017b). At the same time, urban green spaces are becoming increasingly appreciated for the variety of ecosystem services they provide (e.g., Bolund & Hunhammar, 1999; Pauleit et al., 2019). The quality and quantity of the services are often directly correlated with a higher level of biodiversity (Niemelä et al., 2010; Belaire et al., 2022). Urban grasslands are one of the most common types of urban green space in the world (Ignatieva et al., 2020), and they have recently begun to be managed at low intensity to improve the range of ecosystem services they deliver (Chollet et al., 2018; Ignatieva et al., 2020; Przybysz et al., 2021).

In cities, naturally regenerating plant populations are essential for sustaining both the ecological functions and the ecosystem services of urban green spaces (Piana et al., 2019). However, multiple anthropogenic stressors directly and indirectly influence plant reproduction, establishment, and survival and thus limit plant recruitment in the city (Piana et al., 2019). The urban environment is very specific and creates a so-called cityscape (Andersson, 2006) with a high level of habitat fragmentation and a small population of wildlife (Andrieu et al., 2009; Beninde et al., 2016; Hejkal, Buttschardt & Klaus, 2017; Aronson et al., 2017a; Aronson et al., 2017b). The fragmentation strongly influences the maintenance of genetic diversity, the ability to avoid inbreeding, the extent to which populations can respond to divergent selection pressures, and the stability of meta-population dynamics (Allendorf & Luikart, 2007; Emel et al., 2021). In addition to the direct effect of fragmentation, other processes also influence the genetic structure of populations, namely, the founder effect resulting from establishment of new urban populations, and severe bottlenecks due to direct human-related selection pressures, such as herbicide use and soil disturbance (Johnson & Munshi-South, 2017). As a result, urbanisation can change evolutionary processes in cities (Johnson & Munshi-South, 2017; Santangelo James, Ruth Rivkin & Johnson, 2018). The general prediction is that an urban environment leads to reduced genetic diversity within populations and increases diversity among populations (Miles et al., 2019). Although plants can passively disperse between habitat patches via pollen or seeds, such dispersal is only possible when effective connectivity between patches exists (Auffret et al., 2017; Uroy, Ernoult & Mony, 2019). The presence of continuous, narrow habitat fragments serving as corridors between patches, as well as discontinuous or nonlinear habitat fragments serving as stepping stones, can increase landscape connectivity (Van Rossum & Triest, 2012; Uroy, Ernoult & Mony, 2019). Thus, high connectivity between habitat patches might reduce the adverse effects of fragmentation by facilitating genetic flow among local populations. In the case of highly mobile organisms that are well adapted to a city environment, the urban environment can even facilitate gene flow among populations, especially if dispersal is related to human activities. Based on this ‘urban facilitation’ model, there is a weak signature of reduced genetic diversity within populations and no evidence in the literature of a consistent increase in genetic differentiation between populations associated with urbanization (Miles et al., 2019).

The ability of a particular plant species to spread in a fragmented landscape differs in regard to its dispersal strategies, and only some species are able to realize long-range dispersal. Typically, wind dispersal can be a successful mechanism for long-distance dispersal, while gravity-dispersed plants usually spread in a short-range. Dispersal by animals can also results in a dispersal range from less than 100 m in the case of insects and small mammals, up to several kilometres in case of large or migratory animals. However, these vectors are also influenced by the landscape structure, which may in turn influence the plant’s response to decreasing connectivity (for review see Uroy, Ernoult & Mony, 2019). The genetic response to isolation of plant populations also depends on plant biology, with self-incompatible, insect-pollinated, long-lived plants with high colonization ability being better adapted to isolation (Evju et al., 2015; Duwe et al., 2018). Owing to the importance of spatial structures on population genetical processes different approaches for modelling have been employed. Initially, the simple geographic distance (isolation by distance) was used to evaluate the spatial effect; however, it was gradually realized over time that landscape features could substantially affect gene flow patterns, and isolation by resistance has been found to be more realistic (for review see Cruzan & Hendrickson, 2020). Isolation by resistance represents an integration of factors, such as the willingness of an organism to cross a particular environment and the physiological cost and the reduction in survival of doing so. Resistance surface is calculated based on resistance values for particular land cover types, whereby a low resistance denotes ease of movement and a high resistance denotes restricted movement or is used to represent an absolute barrier to movement (Zeller, McGarigal & Whiteley, 2012; Peterman et al., 2019).

The problem of landscape-scale connectivity and the genetic structure of plant populations has been intensively studied in recent years because of its importance for basic science, as well as conservation practice (e.g.,  Auffret et al., 2017; Aavik & Helm, 2018; Hilty et al., 2020). Nonetheless, most research in this area is performed in rural areas, and data from urban areas are less common (Martin, Blossey & Ellis, 2012; La Point et al., 2015). Previous studies (Hejkal, Buttschardt & Klaus, 2017; Mollashahi, Szymura & Szymura, 2020) based on the spatial distribution and size of patches show that urban grasslands have high fragmentation and low connectivity. The spatial structure suggests that rivers crossing a city can be barriers to connecting opposite riverbanks (Mollashahi, Szymura & Szymura, 2020). However, for living organisms, the structural connectivity that emerges from the spatial structure of suitable habitats is less important than the functional connectivity (La Point et al., 2015; Kimberley et al., 2021). For example, waterbodies are assumed to be almost insurmountable barriers to cattle movement (Kimberley et al., 2021), and a medium-size river was found to be able to separate lizard (Podarcis muralis) populations in a city (Beninde et al., 2016). Thus, it seems that a medium-size river can impede long-distance seed transport. Although much of the gene transfer in plants is realized through pollen (Petit et al., 2005), the potential of rivers to impede pollinator movements to opposite banks is poorly studied (but see Zurbuchen et al., 2010).

In this study, we analysed the genetic structure of red clover (Trifolium pratense L.) populations in a cityscape of a medium-size city located in Central Europe. T. pratense is not considered to be a valuable plant for urban grasslands (e.g., Turgeon, 2005), and therefore, we assumed that its population structure mainly reflects spontaneous succession and survival. We sought to answer basic questions regarding the genetic structure of T. pratense populations within a cityscape: Does genetic diversity between populations exceed the genetic diversity within individual populations? Is there a spatial pattern of genetic diversity of T. pratense, such as differentiation between opposite river banks? Does the isolation by distance and/or resistance influence the genetic diversity between populations? Is the size of plant populations and landscape connectivity correlated with the genetic diversity within populations? We hypothesized that the genetic structure of populations is shaped mostly by isolation in the cityscape. However, we also recognized that pollen flow in insect-pollinated plants can maintain the connectivity between populations, thereby allowing small urban grassland patches to serve as stepping stones for pollinators.

Materials and Methods

Study area

The study was conducted in Wrocław, Poland, Central Europe (Fig. 1). Wrocław is located in the Odra River valley at an altitude ranging from 105 to 156 m a.s.l. The total area of the city is about 300 km2, and the city’s population is approximately 650,000. The city is surrounded by a relatively uniform landscape of intensively used agricultural areas and narrow strips of riparian forest and semi-natural vegetation along watercourses. The system of urban green areas consists of urban forests, parks, allotment gardens, and urban grasslands associated with road verges, parks, and lawns. Results of previous studies suggest strong fragmentation of urban grassland patches in the city, including the effect of the Odra River separating the cityscape into north and south parts and impeding seed transfer between them (Mollashahi, Szymura & Szymura, 2020).

Biology of the study species

Red clover (Trifolium pratense L.) is an insect-pollinated crop native to southeastern Europe and Asia Minor. It is a perennial plant belonging to the Fabaceae family, and as a legume it has the ability to fix atmospheric nitrogen through symbiosis with the bacteria Rhizobium leguminosarum biovar trifolii (Taylor & Quesenberry, 1996). It naturally occurs as diploid species (2n = 14), but also exists as tetraploid varieties (2n = 28) in commercial production (Boller, Schubiger & Kölliker, 2010). Red clover is adapted to a wide range of environmental conditions (Taylor & Quesenberry, 1996). Natural populations occur on semi-natural grasslands and old fields, and along roadsides. Additionally, it is one of the most important forage legumes in temperate climates: it is widely used for grass-clover leys in crop rotation, and it is an important component of permanent pastures and meadows (Koelliker, Enkerli & Widmer, 2006).

T. pratense is an obligate out-crossing species with strong self-incompatibility (Boller, Schubiger & Kölliker, 2010; Vleugels et al., 2019). The genetic diversity of natural and cultivated populations of red clover has previously been studied using methods involving isoenzymes (Mosjidis et al., 2004), AFLP markers (Collins et al., 2012; Pagnotta et al., 2011), RAPD (Dias et al., 2008b), SSR markers (Dias et al., 2008a; Ahsyee et al., 2014), and genotyping by sequencing (Jones et al., 2020). These studies reported high genetic diversity, with diversity within populations being larger than between populations.

Figure 1 Localization of Wrocław city (A) and the 16 studied populations of Trifolium pratense in Wrocław (B).

Within each population, eight individuals were sampled, but for map legibility, individual plant locations are not shown. The satellite image (https://mapy.geoportal.gov.pl/, CC BY 4.0) is presented in the background.

T. pratense is pollinated by social bees including wild bumblebees (Bombus ssp.) and honeybees (Apis mellifera L.) (Bohart, 1958; Free, 1993), and various pollinator species play important roles in different geographical regions (Vleugels et al., 2019). Bumblebees with long tongues such as B. pascuorum, B. ruderatus, and B. hortorum are the most efficient pollinators of T. pratense flowers, because their tongues are long enough to reach the nectar at the bottom of the corolla tube (Free, 1993; Wermuth & Dupont, 2010). Short-tongued bumblebees such as B. terrestris and B. lucorum are less effective, because their tongues are often too short to reach the nectar through the corolla tube (Free, 1993; Wermuth & Dupont, 2010). Honeybees face the same difficulties with red clover pollination because they also have short tongues (Free, 1993). However, in hot and dry climate regions, the nectar is located sufficiently high in the corolla tube and honeybees can probably access it (Karagic, Jevtic & Terzic, 2010). Honeybees are also observed as pollinators on red clover, when alternative pollen and nectar sources are deficient (Ruszkowski et al., 1980; Free, 1993; Wermuth & Dupont, 2010).

Long-range seed dispersal is related to endozoochory by cattle and wild ungulates, while short-range dispersal is related to ants (myrmecochory) (Taylor & Quesenberry, 1996; Picard et al., 2015). T. pratense provides a good nectar source for pollinators, and it has recently been suggested as a desirable component of seed mixtures for pollinators (Hicks et al., 2016); however, it is not considered a suitable species for urban grasslands (Turgeon, 2005). Its presence is related to high biomass production (Kozłowski, 2012; Sanderson et al., 2013). In Wrocław city, the majority of lawns are established and restored using mixtures containing only grass seed. Recently, seed mixtures for pollinators have sometimes been used, however, and they contain other Trifolium species such as T. incarnatum and T. resupinatum (information from City Greenery Office of Wrocław). Therefore, we assume that most of the T. pratense populations in the city were not intentionally introduced but exist as a remnant of previous land use or were established in the course of spontaneous succession. Nonetheless, we cannot exclude accidental introduction by contaminated seed mixtures.

Plant material collection and molecular investigations

A total of 128 individual Trifolium pratense plants belonging to 16 populations (eight individuals per population) were collected from Wrocław city (Fig. 2). Individual plants were collected at least 10 m apart from one another in order to accurately characterize the genetic variability of the population. Then, the samples were transported to the laboratory and were keep at 4 °C in sealed bags until DNA extraction. The genomic DNA from plants was extracted using Nucleo Spin Plant II extraction kit (Macherey Nagel, Germany) according to the manufacturer’s protocol. After isolation, the quality of the DNA was checked using 1% agarose gel electrophoresis. For analyses, inter-simple sequence repeat (ISSR) markers were chosen owing to their utility and efficiency (Ziketkiewicz, Rafalski & Labuda, 1994; Conte, Cotti & Cristofolini, 2007; Goldman, 2008; Ng & Tan, 2015) and their high informative value based on their reproducibility and polymorphism (Gupta et al, 2021; Tamboli et al., 2018). In total, about 40 ISSR primer pairs were checked, and 8 primer pairs (Table S1) that showed a satisfactory amplification and generated sufficient polymorphic bands were used for analyses. The number of amplified products varied from 4 to 9 within a size range of 100–2,000 bp, depending on the specific primer. PCR amplifications were performed in 15-µL PCR tubes which contained an amplification mixture utilizing DreamTaq polymerase (Thermo Fisher Scientific, Waltham, MA, USA). The composition of the PCR mixture was as follows: DreamTaq reaction buffer containing MgCl2, a 0.2 mM dNTP mix, 1U DreamTaq DNA polymerase, 0.5 mM ISSR primer, and 0.1 mM genomic DNA. Before the amplification was initiated, the gradient method was used to determine the appropriate annealing temperature. Finally, the PCR cycle consisted of an initial denaturation at 95 °C for 8 min, followed by 30 cycles at 95 °C denaturation, 45 s of annealing at the temperature 50–54 ° C depending on the primer (for details see Table S1), and 45 s elongation at 72 °C, with a final extension of 10 min at 72 °C. For the PCR amplifications, a Veriti Thermal Cycler (Life Technologies, Carlsbad, CA, USA) was used. Finally, the PCR ISSR amplification products were separated on a 1% agarose gel with a DNA mass ruler (Thermo Fisher Scientific) and photographed.

Figure 2 Genetic differentiation of populations of red clover (Trifolium pratense).

The different colours represent the frequency of alleles identified as characteristic for particular clusters, following the results of Bayesian clustering for nine populations. The cluster name codes (A–I), are consistent with codes in Fig. 3. The OpenStreet map (https://www.openstreetmap.org; OpenStreetMap contributors, CC BY-SA 2.0) is used in the background.

Data analysis

For analysis of photographs of separation results CLIQS software (Totallab, 2016) was used, with manual correction if necessary. All bands produced by the markers were encoded as a binary matrix and used for further calculations.

POPGENE v. 1.32 (Yeh, Yang & Boyle, 1999) software was used to calculating observed number of alleles, effective number of alleles (Kimura & Crow, 1964), Nei’s gene diversity index (Nei, 1973), Shannon’s information index (Lewontin, 1972), and Nei’s unbiased measures of genetic identity and genetic distance (Nei, 1978). The number of polymorphic bands and the number of private bands found in groups were calculated using FAMD (Schlüter & Harris, 2006).

Bayesian clustering based on an admixture model was calculated using STRUCTURE 2.3.4 software (Pritchard, Stephens & Donnelly, 2000; Evanno, Regnaut & Goudet, 2005; Falush, Stephens & Pritchard, 2003). Values of delta K for different numbers of populations (K), from 2 to 16, were tested with 10 replications per K using 200,000 burn-in iterations followed by 1,000,000 MCMC iterations. The optimal value of K was assessed based on Evanno’s delta K estimation (Evanno, Regnaut & Goudet, 2005) and visualised using CLUMPAK software (Kopelman et al., 2015).

To determine the distribution of genetic variation within and among the populations and to assess genetic differentiation of the main groups of populations AMOVA (analyses of molecular variance) tests were applied using ARLEQUIN 3.5.1 (Excoffier & Lischer, 2010) software with 2,000 permutations. The degree of population subdivision was measured by Fst, Fsc, and Fct fixation indexes based on Weir & Cockerham (1983), Excoffier, Smouse & Quatro (1992). AMOVA was performed for distinguished groups both by Bayesian clustering and for testing the potential effect of isolation by river.

Principal coordinates analysis (PcoA) was performed on a genetic distance matrix generated from the haploid data. The GeneAlex program was used for this purpose, and the genetic distance was calculated following Huff, Peakall & Smouse (1993).

The urban landscape matrix was quantified using the Topographic Objects Database (BDOT10k), which shows the land cover in Poland. The database generally corresponds to a map scale of 1:10,000, and it is regularly updated by local state services. The database is the most comprehensive recent data set regarding land cover, including urban green areas, available in Poland (Feltynowski et al., 2018). The effect of isolation by distance was explored by comparing matrices of genetic distances with Euclidean distances between T. pratense individuals, and then correlating the matrices using the Mantel test. Since the relation was not linear, we applied the Mantel test based on Spearman rank correlation, using the ‘vegan’ package in the R environment (R Core Team, 2021). The isolation by habitats was examined by means of the least cost path (LCP) approach. The LCPs were determined by the ‘leastcostpath’ package (Lewis, 2022) using the ‘from each to each’ algorithm (White & Barber, 2012). The cost surface was prepared in the ‘gdistance’ package (Van Etten, 2017), based on a rasterized BDOT10k map (for details see Table S3 and Fig. S3). Parametrising the resistance surface, we focused on pollen movement by pollinators, particularly bumblebees, excluding seed movement related to large herbivores as being less likely in a city. We assumed no resistance for urban grasslands; low resistance for forests, parks, and allotments; moderate resistance for rivers, dispersed buildings, and arable grounds; and high resistance for built-up areas and communication routes. In short, resistance was assumed to increase with the urbanisation gradient as represented by a high proportion of built-up areas and the communication network (for details see Table S3 and Fig. S3). Information regarding habitat preferences of the pollinators in Wrocław was taken from Michołap et al. (2017) and Sikora & Michołap (2018). The quantification of resistance surface is generally challenging (for reviews, see Zeller, McGarigal & Whiteley, 2012; Peterman et al., 2019), and we applied optimization of resistance surfaces based on genetic data (Peterman, 2018). However, given the likelihood of a lag effect between the recent landscape configuration and the genetic structure of surveyed populations (Epps & Keyghobadi, 2015), especially in a highly variable cityscape, we decided not to optimise the resistance values by comparing them with genetic data. Finally, the length matrix of LCPs was correlated against genetic distance, using the Mantel test based on Spearman rank correlations. The two models of genetic distance, as a linear function of geographical distance and least cost path length, besides the Mantel test, were also checked using log-likelihood test for non-nested models (Vuong, 1989) as well as by comparison of AIC values. All the results were comparable, but since the data reveal a heteroskedasticity (detailed results not shown) we finally presented only the results of the non-parametric Mantel tests.

The effects of the urbanization gradient and the grassland patch size and connectivity on the genetic diversity of populations were tested by correlating the Nei’s genetic diversity with the above-mentioned landscape-scale grassland patch characteristics. As a proxy for measuring urbanisation, we used the percentage of impervious surface (Fortel et al., 2014; Bartlewicz et al., 2015), and following the approach of Bartlewicz et al. (2015), we calculated the percentage within buffers with radii of 0.5, 1 and 2 km around the centroid of the sampled populations. The details of the impervious surface mapping are presented in the Appendix, and the distribution of impervious surfaces within the Wrocław city is presented in Fig. S4. An integrated index of connectivity (IIC) and its components (intra, flux, connector) was chosen as a measure of connectivity. The importance of each grassland patch was assessed by the mean of changes in the IIC index (dIIC), with high values denoting high connectivity, and low values representing low connectivity. The components intra, flux, and connector represented the interpatch connectivity and direct connection with other patches, which functioned as ‘stepping stones’ to other patches, respectively. For details of this method, see Saura & Torné (2009), Saura & Rubio (2010), and Baranyi et al. (2011). The values of dIIIC and grassland path size were previously calculated for different distance thresholds in the studied city, and the detailed computation methodology is presented in Mollashahi, Szymura & Szymura (2020).

Results

The ISSR primers amplified 112 loci across 128 individuals. Analysis of the data indicated that 85 of the 112 loci were polymorphic, which reflects the high allelic diversity of the studied T. pratense populations. The results of the genetic polymorphism analysis for populations are presented in Table 1. Generally, the lowest level of polymorphism was found in population 16, while the highest was in population 7 (Table 1). All the metrics were significantly correlated (Table S2). Private loci were not observed in six populations, and the highest number (three) was found in population number 7.

Table 1 Centroids of location (N, E), and genetic diversity within red clover (Trifolium pratense) populations in Wrocław.

No	N	E	Na1	Ne2	H3	I4	Pl5	Pl [%]6	Pb7	
1	51.07988	17.04937	1.17	1.09	0.05	0.08	19	16.96	0	
2	51.08427	17.02184	1.20	1.09	0.06	0.09	23	20.54	1	
3	51.13583	17.05738	1.17	1.09	0.06	0.08	19	16.96	1	
4	51.10865	17.14161	1.19	1.07	0.05	0.08	21	18.75	1	
5	51.16295	16.92723	1.21	1.09	0.06	0.09	24	21.43	0	
6	51.12082	16.89901	1.21	1.09	0.06	0.10	24	21.43	0	
7	51.05710	17.00945	1.38	1.18	0.12	0.18	43	38.39	3	
8	51.13294	16.98159	1.27	1.15	0.09	0.14	30	26.79	1	
9	51.13615	16.95091	1.16	1.08	0.05	0.08	18	16.07	0	
10	51.14909	17.02034	1.19	1.09	0.06	0.09	21	18.75	1	
11	51.13657	17.08619	1.19	1.08	0.06	0.09	21	18.75	2	
12	51.10239	17.14313	1.15	1.07	0.05	0.07	17	15.18	0	
13	51.15278	16.92557	1.13	1.05	0.04	0.06	14	12.50	2	
14	51.15551	17.13676	1.25	1.12	0.08	0.12	28	25.00	2	
15	51.19174	16.99583	1.13	1.06	0.04	0.06	14	12.50	0	
16	51.14317	17.13329	1.12	1.05	0.032	0.05	13	11.61	1	
Notes.

(1) Na, Observed number of alleles.

(2) Ne, Effective number of alleles (Kimura & Crow, 1964).

(3) H, gene diversity (Nei, 1973).

(4) I, Shannon’s Information index (Lewontin, 1972).

(5) Pl, number of polymorphic loci.

(6) Pl (%), percentage of polymorphic loci (Pl).

(7) Pb, number of private loci.

The results of Bayesian clustering with Evanno’s delta K estimation suggest that the optimal number of clusters is 9 (Fig. S1). The spatial distribution of results for the 9 clusters are plotted in Fig. 2, while results for all analysed number of clusters are shown in Fig. S2.

The AMOVA indicated that 45% of the total variation was attributable to differences among individuals, and it provides a value for the fixation index of all populations Fst = 0.55 (Table 2). The effect of the Odra River as a barrier, which divides the T. pratense populations into two groups and nine groups, as suggested by Bayesian clustering, reveals low differentiation among groups (0.5% of variation explained by division into two groups, and 13.6% for clustering into nine groups) and high differentiation among populations within groups (54.1% for two groups, 44.4% for nine groups). It corresponds with relatively low values of Fct (0.5 for two groups and 13.6 for nine clusters) and high Fsc values (54.1 and 44.1 for 2 and nine groups, respectively) (Table 2).

Table 2 Population genetic structure inferred by analysis of variance (AMOVA).

The calculations take into account the division of all populations into two (possible effect of river) and nine (results of Bayesian clustering) groups.

Source of variation	d.f.	Sum of squares	Percentage of variation	F statistics	
no groups				
Among population
within groups	15	736.36	54.5		
Within population	112	517.17	45.5	Fst = 0.55	
Total	127	1,272.09	100.0		
2 clusters (opposite river banks)				
Among groups	1	52.45	0.5	Fct = 0.005	
Among population within groups	14	683.9	54.1	Fsc = 0.54	
Within populations	110	517.2	45.4	Fst = 0.55	
Total	127	12,532.2	100.0		
9 clusters				
Among groups	8	467.26	13.6	Fct = 0.14	
Among population within groups	7	269.10	41.4	Fsc = 0.47	
Within populations	110	517.17	45.0	Fst = 0.55	
Total	125	1,253.52	100.0		

The effect of differentiation between populations results in clear separation of some populations in PCoA ordination space (Fig. 3). However, the relatively low genetic variation related to spatial clustering is visible in Fig. 2, which shows no obvious geographical pattern of genetic diversity.

The spatial analysis showed that the populations are significantly isolated by distance (Mantel r = 0.12, p = 0.001, Fig. 4A), and even more by environment (Mantel r = 0.17, p = 0.05 Fig. 4B). The relatively low, yet significant values of Mantel correlation coefficients result from the non-linearity of these relationships. The isolation by distance and by the environment increased strongly up to a Euclidean distance of ca. 2 km, and a LCP length of ca. 5 km. Above these thresholds, we did not observe a further increase, and the genetic distance fluctuated around high values, regardless of the distances (Figs. 4A and 4B).

Figure 3 Principal coordinates analysis plot of the genetic diversity between studied individuals belonging to different populations.

The colours used here are consistent with colours for the nine clusters shown in Fig. 3 and Fig. S2.

Figure 4 The relationship between genetic and geographic distances (A), and the genetic distance and least cost patch (LCP) lengths (B) on the background of points density; (C) the relationship between Nei’s gene diversity and patch size, as well as (D).

Graphs (A) and (B) also show the results of Mantel tests (mantel r, p); the lines were fitted using the ‘loess’ algorithm. Note the logarithmic scale on patch size axis in (C).

We found no correlation between UG patch area and the genetic diversity of the population within the patch (Spearman r = −0.101, p = 0.586, Fig. 4C). We also observed no significant correlations between genetic diversity of populations and landscape connectivity of the settled patches (Table S4), as well as urbanization gradients for all examined buffer radius (Fig. 4D, Table S5).

Discussion

We observed high genetic differentiation between populations, which we assume was caused by long-term fragmentation and stochasticity in historical development and land-use system changes in the urban landscape. The city of Wrocław, like other Central European cities, has a rather long and complex history of spatial development that included absorbing and transforming the surrounding rural areas, as well as undertaking reconstruction after war-related devastation (Bińkowska et al., 2013). These factors likely caused fragmentation, founder effects associated with the establishment of new populations, and bottlenecks resulting from anthropogenic disturbances, such as herbicide use and soil disturbance. All of these processes lead to genetic drift in populations (Johnson, Thompson & Saini, 2015; Johnson & Munshi-South, 2017). The processes could be assumed to occur more or less randomly within a cityscape. Therefore, we did not observe any obvious geographic patterns of genetic diversity within the city (e.g., clearly spatially structured clusters of populations). Population diversity was also higher within cluster on the opposite banks of the Odra River, than between them.

Most of the gene flow in plants is related to pollen flow (Petit et al., 2005), and seed dispersal by large mammals seems to be very limited within a cityscape. Therefore, we assumed that most of the genetic connectivity between populations was related to pollen movement by pollinators. The average foraging distances for Bombus pascuorum and B. terrestris, important pollinators of Trifolium pratense, are 124 m and 395 m, respectively (Damgaard, Simonsen & Osborne, 2008). Previous experiments with Primula elatior pollen transportation by insects showed that most transferring events occur over short distances (80% within 15–115 m), with a maximum distance of about 650 m. In the absence of stepping-stone habitat patches, plant populations were separated at a distance of 250 m, while the presence of small stepping-stones increased the interpopulation pollen transfer up to 600 m (Van Rossum & Triest, 2012). Estimated between-population long-distance pollen deposition has also been observed for other insect-pollinated herbs in the same urban context, up to 524 m for Lychnis flos-cuculi, 743 m for Centaurium erythraea, and 2.58 km for Centaurea jacea (Van Rossum, 2009; Van Rossum & Triest, 2010). Our findings seem to reveal a similar pattern: the genetic distance within a patch was relatively low, but it sharply increased up to a distance of 2 km and then fluctuated at a high level. Isolation-by-resistance (McRae, 2006) is better correlated with the genetic differentiation observed here, compared with isolation-by-distance (Wright, 1943; Slatkin, 1993). Thus, the structure of a cityscape significantly restricts functional connectivity between T. pratense populations via a pollinator network. The differences in effects of Euclidian distance and distance along a LCP on the genetic distance between populations support the assumption regarding lower permeability of urbanised areas as communication routes and built-up areas compared with the green infrastructure. As mentioned previously, little is known regarding the movement of pollinators cross medium-size rivers (Zurbuchen et al., 2010). Large rivers, such as the Yangtze, can isolate populations of wild pollinators (Deng et al., 2020). Results obtained in the current study do not suggest that the Odra River, with an average width of slightly more than 100 m, is a barrier that strongly influences pollinator movement.

Populations from larger patches are generally assumed to be more genetically diverse than populations from small patches (Lande & Barrowclough, 1987; Frankham, 1996; Kardos et al., 2021). Similarly, the populations in patches with higher connectivity are assumed to be more diverse than those of isolated patches (Lande & Barrowclough, 1987; Kardos et al., 2021). However, our results are counter to these assumptions. Additionally, it is hypothesised that in more urbanised areas, plant populations may experience higher environmental pressure resulting from factors such as drought and the mowing regime. Such pressure can lead to elimination of less adapted genotypes (Bartlewicz et al., 2015). For example, in the case of clonal plants Linaria vulgaris environmental pressure caused clonal reproduction to dominate at the in expense of sexual reproduction in most urbanized areas (Bartlewicz et al., 2015). However, we did not observe a relationship between urbanisation level and genetic diversity of particular populations. We linked these outcomes to the relatively high dynamics of the landscape structure in the city. In modern times, changes in UG patch area and connectivity have been fairly rapid, and plant populations are often out of genetic equilibrium with the recent landscape structure because genetic structure may represent past, rather than the contemporary landscapes (the time-lag effect reported by Llorens, Ayre & Whelan (2004)). This lag between current demographic processes and population genetic structure often leads to challenges in interpreting how contemporary landscapes and anthropogenic activity shape gene flow (Epps & Keyghobadi, 2015). The lag effect is especially pronounced in long-living, clonal plant species (Honnay et al., 2005; Fuller & Doyle, 2018), such as Trifolium pratense. Typically, historical maps are used to test possible time-lag effects (Epps & Keyghobadi, 2015), but unfortunately, similar to other cities, no detailed historical maps of land cover are available for Wrocław. The lack of correlation between population/UG patch size and genetic differentiation has a consequence for greenery management, underscoring that even small patches have value in conservation programs of urban environment (Soanes et al., 2019).

A few studies have confirmed that urbanization influence plants’ evolutionary processes and patterns (Johnson, Thompson & Saini, 2015). An analysis of adaptive evolution of dispersal traits in urban populations of Crepis sancta (Cheptou et al., 2008) showed that plants within highly-fragmented urban landscape evolved to produce a greater proportion of non-dispersing seeds than plants from surrounding, unfragmented populations. This finding is related to the fact that dispersing seeds have a lower chance of settling in their patch compared with non-dispersing seeds and often fall on a concrete matrix unsuitable for germination. Urban landscapes also influence the genetic structure of Linaria vulgaris populations, which exhibits lower genetic diversity and fitness in urban populations than in rural ones (Bartlewicz et al., 2015). Similar dependency was observed in the case of Impatiens capensis populations, in their native range in Canada (Rivkin & Johnson, 2022). Opposite situation was observed in invasive populations of Ambrosia artemisiifolia in East China, where urban populations had more complex genetic patterns than non-urban populations (Lu et al., 2022). This pattern may have been shaped by the long-distance dispersal of seeds through human activities such as transportation in the urban environment. The adaptive divergence of competitive traits between urban and rural populations was observed in two annual plants Digitaria ciliaris and Eleusine indica, of the Poaceae family in the Tokyo metropolis and the surrounding rural landscape (Fukano, Uchida & Tachiki, 2022). In urban habitats, lower density and weaker competition will lead to reduced competitive ability in plant populations. However, in the case of populations of Trifolium repens, no changes in genetic diversity with increasing urbanization were observed, indicating that genetic drift is unlikely to explain observed urban–rural clines in the frequency of hydrogen cyanide (HCN), which is produced as a potent antiherbivore defense (Johnson et al., 2018).

Limitation of the study

The high diversity within, as well between studied populations is usually observed in Trifolium pratense, owing to the out-crossing character of the species (Boller, Schubiger & Kölliker, 2010; Vleugels et al., 2019; Jones et al., 2020). Consequently, numerous individuals need to be screened to reveal the genetic variability, and capture rare alleles in each population. In this regard, our sample of eight individuals per population was relatively low, but our study focused on analysing the relationships between the genetic structure of red clover populations and isolation caused by distance and resistance in a cityscape. The number of sampled populations was also not high; however, sites were selected to analyse the differences between the city center and peripheries and along main geographic directions, allowing to examination of the effect of the river as a barrier (north-south gradient) and as a corridor along the water course (west-east direction). The values obtained here cannot be directly compared with other results for the Trifolium genus (e.g., Mosjidis et al., 2004; (Dias et al., 2008a; Dias et al., 2008b; Pagnotta et al., 2011; Collins et al., 2012; Ahsyee et al., 2014; Johnson et al., 2018; Jones et al., 2020)), partly because of differences between studied landscapes, but mostly because of differences in genetical analysis methods (e.g., usage of different markers), which can cause differences in the estimation of genetic characteristics in populations.

Conclusions and practical implications

• Great variation exists within and between Trifolium pratense populations within the city, with no discernible geographic pattern of genetic diversity. This finding is likely due to the long history of the city and the high stochasticity of land-use changes along with city development. In particular, we showed that the Odra River, with a width of ca. 100 m, is not a strong barrier for gene transfer in red clover.

• Strong isolation occurs through resistance and distance, which means that the populations are threatened by genetic drift.

• Since naturally regenerating plants are critical to sustaining both the ecological functions and the ecosystem services of urban green areas, gene movement between populations should be increased by appropriate management of urban green areas.

• Small UG patches, with small populations, can still hold rare alleles which significantly contribute to the overall genetic variation of Trifolium pratense in a city. This observation supports the importance of maintaining and preserving UG patches as much as possible, irrespective of their size.

Supplemental Information

Supplemental Information 1 Raw data of sequencing

Click here for additional data file.

Supplemental Information 2 Supplementary figures and tables

Click here for additional data file.

Additional Information and Declarations

Competing Interests

Author Contributions

Data Availability

The authors declare there are no competing interests.

Hassanali Mollashahi conceived and designed the experiments, performed the experiments, analyzed the data, prepared figures and/or tables, authored or reviewed drafts of the article, and approved the final draft.

Jacek Urbaniak conceived and designed the experiments, performed the experiments, analyzed the data, prepared figures and/or tables, authored or reviewed drafts of the article, and approved the final draft.

Tomasz H Szymura conceived and designed the experiments, analyzed the data, prepared figures and/or tables, authored or reviewed drafts of the article, and approved the final draft.

Magdalena Szymura conceived and designed the experiments, analyzed the data, prepared figures and/or tables, authored or reviewed drafts of the article, and approved the final draft.

The following information was supplied regarding data availability:

The raw data are available in the Supplemental Files.

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
