# Peer review of "Genetic structure of Trifolium pratense populations in a cityscape"

_PeerJ, doi:10.7717/peerj.15927_

## Round 0.1 · original submission · Major Revisions

Dear Dr. Szymura,

Please answer all points raised by the reviewers.

Reviewer 1 ·

Basic reporting

The research appears well-perfomed and the paper is well-written.

Overall, the English is of high quality. I suggest only a handful of minor textual edits.
Sufficient background is provided. Some minor improvements are possible here.
Studture, tables, figures and raw data are OK.
The paper describes a coherent, self-contained work.

Experimental design

All OK.

Validity of the findings

All OK.

Additional comments

Dear reviewers,
I believe this review holds important information. The research is well-performed and well-presented. Some minor modifications need to be carried out before the paper can be published.

General comments:
- 16 populations were sampled with 8 indivuduals per population. Red clover is an outcrossing species. Intra-population genetic diversity is typically very high: much higher than can be captured with 8 individuals. To get a more or less complete idea of genetic diversity within red clover populations, en to capture rare alleles in populations, at least 30 individuals per population need to be screened. This was not done here, but assessing intra-population diversity was not the scope. For the current study in which populations are compared, I believe 8 indivuduals could suffice. Nonetheless, it is important to mention this in the discussion and /or introduction.
- The results of the Structure analysis are only briefly displayed in the paper. I expected to see a plot or table with the results per K value.

Minor remarks:
- line 114: 'isolation by resistance': not clear. Please explain in the preceding text.
- Line 141 - 143: Pollination of red clover needs to be better explained. Nectar is only accessible for some pollinators, mostly bumblebees with long tongues. I suggest to read and refer to the following paper: "Vleugels et al. 2019, Factors Underlying Seed Yield in Red Clover: Review of Current Knowledge and Perspectives"
- line 181: 'the right temperature': more information is needed on the exact annealing temperatures. This could be described in a table when annealing temperatures differ between primer pairs. You can refer to Table S1
- line 195: 'numbers of K': you need to describe that K stands for the number of subpopulations
- Line 273: use same amount of digits after comma for r values. R = 0.119 --> R = 0.12
- Line 313: "isolation by resistance": this temr needs to be introduced in the introduction. Many readers will not know it.
- Line 321-322: sentence needs to be re-written. SOmething like this. "A generally assumed idea is that populations from larger patched are more genetically diverse". Your results prove the opposite in red clover.
- Line 328-332: time lag effect is not clear. Explain in the text please
- Table 1: reduce the numbers after the comma for columns Na, Ne, H, I, to 2. This will increase readibility of the table.
- Table 2: columns "d.f.", "Sum of squares" and "Variance components" can be dropped.
- Additional textual suggestions ar egiven in the attached document.

Annotated reviews are not available for download in order to protect the identity of reviewers who chose to remain anonymous.

Reviewer 2 ·

Basic reporting

I have completed a review of MS 82645, "Genetic structure of Trifolium pratense" populations in a cityscape. In this paper the authors seek to understand how genetic variation within and population populations of T. pratense is related to urbanization. There is substantial genetic variation within populations and substantial genetic structure among populations. The genetic differentiation of populations was not clearly related to urbanization or a natural barrier (a river). They detected isolation by distance, and the Mantel's r was slightly higher for a model of isolation by resistance.

Overall I thought the questions, and results were interesting. The paper is novel in that relatively few studies have looked at how urbanization affects plant genetic variation, although the paper does a relatively poor job of highlighting these previous studies. The strength of the paper could be improved with some further analysis, and some of the writing would benefit from further revision by a fluent English speaker/copy-editor.

Experimental design

There were relatively few populations sampled (N=16), relatively few individuals per population (N=8) given the low number of populations, and the use of 8 ISSR markers gives limited insight into genome-wide patterns of diversity. Ideally the number of populations would have been 2-3x higher, which would have given a better picture of landscape genetic patterns. If they had increased the number of populations they would have been able to decrease the number of samples within populations. However, this is a moot point because I do not think it something that can be easily corrected and I wouldn't advocate for the authors to expand sampling just to make the paper stronger.

This study also focuses on just a single city, so how these results generalize across cities is unclear. While there has been a push to increase the number of cities sampled in many urban evolution studies, it is still the case that most research focuses on a single city, so I cannot fault the authors too much on this aspect of the paper.

Validity of the findings

The caveats mentioned above notwithstanding, the population genetic and landscape genetic analysis seem to have been well done.

I would like to see some further analysis on the following:
- How do measures of urbanization (e.g. % impervious surface, NDVI etc) relate to the amount of genetic diversity within populations?
- How do differences in urbanization (e.g. differences in % impervious surface) relate to the genetic distance between populations? This can be done using a Mantel test or Redundancy Analysis, which some would argue is more accurate than Mantel tests.

- It is unclear whether the isolation by resistance model is really a better fit than isolation by distance. The r values are only slightly different, and no comparison is made between models using AIC or a log-likelihood ratio test. I would be cautious with the interpretation here, and maybe indicate that you have not been able to distinguish which is a better fit with any level of confidence. My guess is these two models are statistically equivalent.

Additional comments

The authors should add a paragraph in the discussion comparing their results with what others have found in plants. I suggest looking to the results out of the labs of PO Cheptou, P. Tiffin, M. Johnson, Bronte.
Here are some other papers I found in a quick search too:
https://www.sciencedirect.com/science/article/pii/S2351989422002165
https://link.springer.com/article/10.1007/s10682-022-10215-3
There are others...
For example, the comparison with the extensive work in white clover seems like a natural fit here, since these plants are congeners.

---

## Round 0.2 · Minor Revisions

Please address the comments raised by the reviewers and resubmit your valuable manuscript again.

Reviewer 1 ·

Basic reporting

OK

Experimental design

OK

Validity of the findings

OK

Additional comments

Dear reviewers,

I agree with the changes made to the manuscript. I only have a few more minor comments before the manuscript can be accepted.


line 167: "but also exists as tetraploid varieties". In fact, more than half of the commercial varieties are diploid. Tetraploid varieties are certainly noy the majority.

line 186: 'and' instead of 'but'

line 195 : 'can probably access it'. honeybees often forage red clover for pollen only.

line 459: 'reveal the' instead of 'show'

line 459: 'In this regard, our sample ...'

Reviewer 2 ·

Basic reporting

I was reviewer 2 on the first round and I have reviewed the revisions made to the latest version.

The authors have addressed most of my earlier comments, which I think have made the paper stronger and clearer. I have only a few minor additional comments:

Reviewer 1 suggested some information be removed from table 2, namely the df, SS and variance components. Information on df, SS should be added back in because this helps readers familiar with statistics evaluate whether the models were properly specified. However, “variance component” should be defined more clearly, and I am personally confused about that term here. Variance component is typically used when running random effects models. Typically authors will include means square error or F-values, but the variance component does not apply to that, so some clarification is needed on what this values is and how it was calculated - I was not able to reproduce the number myself from the numbers in the table.

The authors are resistant to including additional results support the isolation by resistance conclusion. I think most readers will be skeptical of this result, but I am more convinced after reading the authors response to my constructive criticism. As a compromise, I suggest the authors include something in the methods that says: "Additional analyses comparing AIC values and non-parametric tests (name them XXX) further support our Mantel tests so we only report those results from the Mantel test". That will hopefully assuage concerns of other skeptical critical readers.

At the end of the discussion the authors write: "The values obtained here cannot be directly compared with other results for the Trifolium genus, ...". I don't completely buy their arguments, but at the very least the authors should cite what other results they are referring to.

Experimental design

NA

Validity of the findings

NA

Additional comments

NA

---

## Round 0.3 · accepted · Accept

Please address any additional editorial requests from the editorial office.